# Trends in Calcium Intake among the US Population: Results from the NHANES (1999–2018)

**DOI:** 10.3390/nu16050726

**Published:** 2024-03-02

**Authors:** Zhongyi Yu, Yaqi Li, Djibril M. Ba, Susan J. Veldheer, Liang Sun, Tingting Geng, Xiang Gao

**Affiliations:** 1Department of Nutrition and Food Hygiene, Institute of Nutrition, School of Public Health, Fudan University, Shanghai 200032, China; zhongyi_yu@fudan.edu.cn (Z.Y.); yaqi_li@fudan.edu.cn (Y.L.); sun_liang@fudan.edu.cn (L.S.); 2Department of Public Health Sciences, Penn State College of Medicine, Hershey, PA 17033, USA; djibrilba@pennstatehealth.psu.edu; 3Department of Family and Community Medicine, Penn State College of Medicine, Hershey, PA 17033, USA; sjv152@psu.edu; 4Department of Epidemiology and Biostatistics, School of Public Health, Tongji Medical College, Huazhong University of Science and Technology, Wuhan 430030, China

**Keywords:** calcium, trend, diet, dietary supplement

## Abstract

Inadequate calcium intake is common in the US. Trends in calcium intake among the US population have been less studied, especially in more recent years. We used data from the National Health and Nutrition Examination Survey (NHANES) 1999–2000 to 2017–2018 to study trends in calcium derived from diet and dietary supplements among the US population aged 2 years, stratified by sex, age group, race, and ethnicity. Among the 80,880 participants included in our study, a substantial portion consistently lacked sufficient calcium intake, even when considering calcium from supplements. Concerning trends were observed over the more recent ten years (2009–2018), with decreased dietary calcium intake and no significant improvement in the prevalence of dietary calcium intake < Estimated Average Requirement (EAR) or the prevalence of taking calcium-containing dietary supplements among them. Decreasing trends in dietary calcium intake were more concerning among men, children, and non-Hispanic Whites. Attention should be given to subgroups with higher calcium intake requirements (e.g., 9–18 years and 60+ years), and subgroups with low levels of dietary calcium and a low prevalence of obtaining calcium from dietary supplements (e.g., the non-Hispanic Black subgroup). Concerning trends of calcium intake were observed among the US population from 2009 to 2018. Tailored guidance on dietary choices and dietary supplement use is required to change consumers’ behaviors.

## 1. Introduction

Calcium plays a crucial role in many aspects of human health, including the formation and metabolism of bones, the activation of clotting factors, and the regulation of blood pressure and nerve functions [1,2,3,4,5,6]. Dietary calcium deficiency is a worldwide public health concern [7,8,9,10,11]. In the United States (US), it was reported that 42% of Americans failed to reach their estimated average requirements (EARs) for calcium, based on data from the National Health and Nutrition Examination Survey (NHANES) 2009–2010 [12]. Calcium was classified as one of the four dietary components of public health concern for the general US population [13]. 

While the insufficient intake of calcium has been emphasized in many previous studies [12,13,14,15,16,17], the trends of calcium intake and hence whether such a concerning situation has been improved are less studied. It was briefly mentioned in Dietary Data Brief No. 13 (2014) (https://www.ncbi.nlm.nih.gov/books/NBK589560/, accessed on 2 May 2023) that the dietary calcium intake among the US population increased to around 85 to 190 mg from 1994 to 2010, but no more detail was reported [12]. Previous reports on the trends observed in the consumption of ultra-processed food [18,19], water and beverages [20,21,22,23,24,25], and the intake of added sugar [26,27] have suggested changes in dietary choices among the US population over the past two decades. An assessment based on the NHANES 2003–2006 showed that dietary supplement use among the US population was not sufficient in helping them meet their recommended calcium intake levels [17]. Previous studies suggested that the prevalence of dietary supplement use among the US population increased from 1986 to 2018 [28,29,30]. Supplemental calcium intake was reported to increase from 1999 to 2007–2010 with a subsequent decrease until 2014, but the evaluation did not include supplemental and dietary intake together [31]. Consequently, it was unclear how many individuals with insufficient dietary calcium intake used supplements. 

To address these research gaps, we aimed to assess the trends in calcium intake, with and without dietary supplements, over 20 years from 1999–2000 to 2017–2018 in the US population, using data from the NHANES. Data on such past trends among the population and subgroups of particular interest would be a critical prerequisite for policymakers to make scientific and successful communication strategies that could lead to behavioral changes to move toward healthier dietary patterns.

## 2. Materials and Methods

### 2.1. Data Source and Study Population

The NHANES is a repetitive cross-sectional survey with a complex, multistage probability sample design. It is conducted by the National Center for Health Statistics (NCHS) to obtain health-related information about the civilian, noninstitutionalized population in the US. Details of the study design, protocol, and data collection were described elsewhere [32]. The protocols were approved by the NCHS Ethics Review Board. For individuals aged 2–17 years old, parental/guardian written informed consent was obtained [33,34,35].

In the current study, we included individuals aged two years or older who had complete and valid data on the first-day 24-hour dietary recall during the ten cycles of the NHANES between 1999–2000 and 2017–2018. Previous studies showed disparity in dietary patterns by sex, age, race, and ethnicity [14,36,37,38,39]. Therefore, the potential differences in dietary patterns among subpopulations with different sociodemographic factors were investigated. To allow for tailored communication strategies, we examined the data overall and by age group, sex, race, and ethnicity. Considering that different dietary recommendations are given to people at different life stages, we defined the age groups according to *Dietary Guidelines for Americans 2020–2025* [13], with modification: Children aged 2–18 years old were divided into the age groups of 2–8 years old and 9–18 years old according to previous studies [26,40]. This resulted in four age groups: children aged 2–8 years old, children aged 9–18 years old, adults aged 19–59 years old, and older adults aged 60 years old or above. To reduce the impact of extreme records, the 82,534 sample persons were divided into eight subgroups by sex and age group. The top and bottom 1% for energy intake among each of the subgroups were further excluded from this study, resulting in 80,880 sample persons in total. The overall response rate was 86.2% (see Figure A1).

### 2.2. Dietary Calcium Intake Assessment

Dietary intake data and data on the calcium derived from individual foods were obtained from the 24 h dietary recall survey data “Dietary Interview—Individual Foods”. For survey cycles from 2003–2004 to 2017–2018, when two 24-h dietary recalls were performed, only the first day’s 24-h dietary recall data was included in this study to keep data consistent across all survey cycles [19,41,42]. Each sample person was assigned his/her EAR, recommended dietary allowance (RDA), and tolerable upper intake level (UL), based on his/her sex, age, and pregnancy and lactation status, according to dietary reference intakes (DRIs) determined by Institute of Medicine of the National Academies [43] (see Table A1). Only sample persons clearly self-reported to be pregnant or breastfeeding a child were assigned pregnant/lactating DRIs. Those whose responses were reported as “refused”, “don’t know” or “missing” were assumed to be not pregnant/breastfeeding. The dietary and total calcium intake of each sample person was compared with his/her DRIs to classify the sample person’s diet as “<EAR”, “≥EAR, <RDA”, “≥RDA, ≤UL”, or “>UL”. Seven subpopulations with different DRIs were defined based on age and sex: children aged 2–3 years old, 4–8 years old, and 9–18 years old; adults aged 19–50 years old; men aged 51–70 years old; men aged above 70 years old; and women aged above 50 years old.

### 2.3. Dietary Supplementation of Calcium

Each sample person was asked to report any dietary supplement taken within the last 30 days. For each dietary supplement reported, its ingredient list was screened for the keyword “calcium”. Any supplement with any ingredient name with “calcium” (e.g., calcium, calcium phosphate, etc.) was considered a calcium-containing dietary supplement. This was then used to identify sample persons whose dietary calcium intake did not reach their EARs and who did not take calcium-containing supplements (“<EAR, no supplement” group).

From the 2007–2008 survey cycle and onward, data were available on the total calcium derived from dietary supplements on the day of the 24 h dietary recall. The total calcium intake (dietary and dietary supplement) was calculated and compared to DRIs to identify populations whose calcium intake was still insufficient, as well as those exceeding their ULs.

### 2.4. Statistical Analysis

All analyses were conducted according to the NHANES analytical guidelines, which include sampling weights, stratification, and clustering, to account for the complex NHANES sampling design and ensure nationally representative estimates [18,19,20,44]. For each survey cycle, the weighted percentage of the population falling into different categories (such as the categories established by comparing their intake with their DRIs) with 95% confidence intervals were calculated using Proc Surveyfreq (SAS Institute). The weighted mean daily calcium intake, quartiles, and their respective 95% confidence intervals were calculated using Proc Surveymeans (SAS Institute). The absolute difference was calculated by comparing the weighted mean daily calcium intake from 1999–2000 to 2017–2018. The *p* values for trends were estimated by making the survey cycle a continuous variable and testing it with a linear regression model using Proc Surveyreg (SAS Institute). The speed of the increasing/decreasing trend of calcium intake was presented using the point estimate with 95% confidence intervals of the coefficient of the linear regression and converted to mg/10 years. Day-one dietary sample weight was used for calculating the weighted mean calcium intake and the proportions of the population of each survey cycle. Appropriate weights for the combined NHANES survey cycles were used based on the NHANES analytic guidelines to calculate *p* values for trends over multiple survey cycles [45].

All analyses were conducted with SAS version 9.4 (SAS Institute Inc., Cary, NC, USA), and two-tailed *p* < 0.05 was considered statistically significant. Given the potential for type I error due to multiple comparisons, outcomes for stratified analyses should be interpreted as exploratory.

## 3. Results

After excluding 11,325 participants without valid first-day 24 h dietary recall and 1654 participants with extreme energy intake (those falling under the top and bottom 1% energy intake within each sex and age group), data from a total of 80,880 individuals were analyzed. There were 12,768 participants aged 2–8 years old, 19,196 aged 9–18 years old, 34,043 aged 19–59 years old, and 16,527 aged 60 years or above. Details of the unweighted sample sizes in each survey cycle and among each subgroup are listed in Table 1 and Figure A1.

### 3.1. Dietary Calcium Intake

The mean daily dietary calcium intake increased from 848 ± 28.6 mg/d in 1999–2000 to 1022 ± 13.7 mg/d in 2009–2010 but then slightly dropped to 966 ± 24.2 mg/d in 2017–2018 (Table 2, Figure 1). Both the increase in 1999–2008 (118 ± 56.4 mg/d·10 years^−1^) and the decrease in 2009–2018 (−83.2 ± 36.3 mg/d·10 years^−1^) were statistically significant, with an overall significantly increasing trend observed in the 1999–2018 period (59.0 ± 15.9 mg/d·10 years^−1^) (Table 3).

When stratified by sex, age group, race, and ethnicity, men, children aged 9–18 years old, non-Hispanic Whites, and Mexican Americans had relatively higher dietary calcium intake. Across the ten survey cycles, women, adults aged 60+ years, and non-Hispanic Blacks consistently exhibited the lowest dietary calcium intake within their demographic subgroups. Among all age groups, adults aged 60+ years experienced the fastest increase in dietary calcium intake during 1999–2008 (136 ± 66.4 mg/d·10 years^−1^), followed by the slowest decrease in 2009–2018 (−44.9 ± 67.6 mg/d·10 years^−1^). The most significant decrease in 2009–2018 was observed among children aged 9–18 years (−135 ± 91.0 mg/d·10 years^−1^), followed by children aged 2–8 years (−130 ± 61.0 mg/d·10 years^−1^). From 1999 to 2008, the increase in dietary calcium intake was most significant among other Hispanic (183 ± 85.8 mg/d·10 years^−1^) and non-Hispanic Black (182 ± 57.9 mg/d·10 years^−1^) groups, followed by Mexican American (105 ± 73.9 mg/d·10 years^−1^) and non-Hispanic White (104 ± 77.3 mg/d·10 years^−1^) populations. A similar but not statistically significant increase was observed in other races and ethnicities (123 ± 175 mg/d·10 years^−1^). From 2009 to 2018, a significant decrease in dietary calcium intake was only observed among non-Hispanic Whites (−111 ± 46.2 mg/d·10 years^−1^) across all the races and ethnicities tested.

### 3.2. Proportion with Insufficient or Excessive Dietary Calcium Intake

From 1999 to 2018, the proportion of the US population (≥2 years old) not reaching their calcium EAR with their diet dropped from 60.1 ± 3.45% in 1999–2000 to 48.1 ± 2.60% in 2009–2010 but then remained stable until 49.8 ± 2.60% in 2017–2018 (Figure 1, for detailed data, see Appendix A). Similar trends were observed among most subpopulations when stratified by sex, age group, race, and ethnicity. Women, children aged 9–18 years old, adults aged 60+ years old, other Hispanics, non-Hispanic Blacks, and other races and ethnicities consistently had over 50% of the subpopulation with insufficient dietary calcium intake over the ten survey cycles tested. We did not observe notable improvement in the prevalence of dietary calcium less than EARs in any strata tested (by sex, age group, race, and ethnicity) since 2009–2010 (Appendix A). A small proportion of the population had dietary calcium above UL, fluctuating from 1.34 ± 0.35% in 1999–2000 to 3.48 ± 0.83% in 2009–2010, then to 2.54 ± 0.67% in 2017–2018. The quartiles of dietary calcium intake among subpopulations with different DRIs are presented in Figure A2, according to which children aged 9–18 years old, men aged above 70 years old, and women aged above 50 years old consistently had median dietary calcium intake below their EARs.

Among those whose dietary calcium intake failed to reach their EARs, the proportion of those with any calcium-containing dietary supplement taken within the past 30 days slightly increased from 37.0 ± 3.91% in 1999–2000 to 43.7 ± 2.79% in 2005–2006 but then declined to 33.3 ± 2.39% in 2017–2018 (Appendix A). This prevalence was lower among men, children aged 2–8 years old and 9–18 years old, Mexican Americans, other Hispanics, and non-Hispanic Blacks. Men, adults aged 60+ years, and non-Hispanic Whites consistently had the highest rate of dietary supplement use within their demographic subgroups.

### 3.3. Total Daily Calcium Intake

When the calcium intakes derived from diet and dietary supplements on the same day were combined (from 2007–2008 to 2017–2018 due to data availability), the overall daily calcium intake increased from 1081 ± 55.3 mg/d in 2007–2008 to 1160 ± 16.5 mg/d in 2009–2010 and then decreased to 1061 ± 25.5 mg/d in 2017–2018 (Table 2). Similar decreasing trends were observed among most strata, with the most significant decrease in 2009–2018 observed among non-Hispanic Whites (−170 ± 49.7 mg/d·10 years^−1^). Men, adults aged 60+ years, and non-Hispanic Whites consistently had the highest calcium intake, while women, children aged 2–8 years, and non-Hispanic Blacks consistently had the lowest calcium intake among their demographic subgroups. The proportion of the US population whose daily calcium intake did not reach their EARs dropped from 43.7 ± 3.93% in 2007–2008 to 39.7 ± 1.29% in 2009–2010 but then increased again to 44.1 ± 2.45% in 2017–2018 (Figure 2, Appendix A). Among the four age groups included in this study, children aged 9–18 years old consistently had the highest proportion with total calcium intake (from diet and dietary supplements) lower than their EARs, followed by adults aged 60+ years. Non-Hispanic Blacks, followed by other races and ethnicities, had the highest proportion of participants with insufficient calcium intake among the four racial groups studied. This proportion was consistently the lowest among men, children aged 2–8 years, and non-Hispanic Whites among all demographic subgroups across the ten survey cycles. The proportion of those who reached their UL fluctuated between 4.69 ± 1.18% in 2015–2016 and 6.99 ± 0.89% in 2009–2010. The quartiles of total calcium intake among subpopulations with different DRIs are presented in Figure A3.

## 4. Discussion

In this nationally representative sample of the US population with data from 1999 to 2018, a consistently large proportion of the population had insufficient dietary calcium intake, even after including the calcium derived from dietary supplements. From 1999 to 2018, the dietary calcium intake level increased but then decreased since it peaked in 2009–2010. The trends observed over the more recent ten-year period (2009–2018) are concerning: (1) no significant improvement was observed in the prevalence of dietary calcium intake < EAR; (2) the mean dietary calcium intake decreased significantly; and (3) no improvement was observed in the prevalence of calcium-containing dietary supplement use among those whose dietary calcium intake failed to reach their EARs.

Similar trends were observed among most of the strata tested (by age groups, sex, race, and ethnicity). The increasing trend from 1999–2000 to 2007–2008 was significant among all the strata tested, except for other races and ethnicities. Compared to men, women had relatively lower dietary calcium intake but were more likely to take calcium-containing dietary supplements when their dietary calcium intake alone was not sufficient for them to meet their EARs. Among the four age groups, the decreasing trends in calcium consumption from 2009 to 2018 were most notable among children (both aged 9–18 years old and 2–8 years old), which should be brought to attention. Although children aged 9–18 years old had relatively higher dietary calcium intake, they had the highest proportion of participants with insufficient calcium intake due to their high calcium requirements and lower rates of calcium-containing dietary supplement use. Adults aged 19–59 years old had the fastest increase from 1999 to 2008 and the slowest decrease from 2009 to 2018. However, this age group still consistently had the lowest dietary calcium intake. This age group also consistently had the highest rate of calcium-containing dietary supplement use, making them the subgroup with the highest values in terms of total calcium intake. However, this subgroup still had the second highest rate of insufficient calcium intake due to their higher EARs.

Non-Hispanic Blacks, followed by other races and ethnicities, consistently had the highest proportion of participants with insufficient calcium intake (from diet and dietary supplements) from 2007 to 2018. Non-Hispanic Blacks not only had low dietary calcium intake but also had a lower proportion of calcium-containing dietary supplement use among those with insufficient dietary calcium intake. Both Mexican Americans and non-Hispanic Whites had relatively higher dietary calcium intake across all racial groups, together with similar increasing rates in 1999–2008. However, dietary calcium intake significantly decreased in 2009–2018 among non-Hispanic Whites but not in other racial groups. Compared to non-Hispanic Whites, Mexican Americans were less likely to obtain additional calcium from dietary supplements when their dietary calcium intake was insufficient, which explained their higher rate of insufficient total calcium intake. Among the racial groups tested, the sharp decrease observed in dietary calcium intake from 2009–2010 to 2017–2018 among non-Hispanic Whites should also be brought to attention.

While concerns regarding calcium deficiency among the US population have been clearly established previously [12,13,14,15,16,17], the results reported in this study reveal a concerning decline in calcium intake among the US population since the 2009–2010 survey cycle, immediately after an increasing trend was reported according to a previous study [12]. This highlights the urgent need for effective measures to improve the dietary patterns of those who fail to reach their calcium EARs and to encourage the use of calcium-containing dietary supplements among those needed. The stratified analysis revealed different patterns across each demographic subgroup, which helps to identify those subgroups in need of better guidance on their food choices and/or dietary supplement use. This includes subpopulations not making enough effort to fulfill their increased need for calcium at critical life stages (such as children aged 9–18 years old), subpopulations with relatively lower proportions not reaching their EARs but showing concerning trends that need behavior adjustment (such as children aged 2–8 years old), and subpopulations with consistently lower proportion meeting their EARs who need urgent interventions (such as the non-Hispanic Black subgroup). This helps policymakers in making tailored messages and measures for targeted subgroups that might be more successful in changing people’s perceptions and behaviors [46,47,48]. For example, messages given at certain events or places are more likely to reach certain subpopulations of interest. Social media has made it easier to push relevant messages to specific audiences. Calcium intake from diet and dietary supplements might be influenced by multiple factors such as culture, food preferences, health status, family income, education, environment, knowledge of nutrition science, and perception of certain foods and dietary supplements. Further analyses of trends in calcium-rich foods and the influence of potential factors are required to reveal the underlying changes in dietary choices that led to the trends observed in this study. Those who have difficulty in reaching their EARs with dietary calcium intake alone should be identified to support better guidance on dietary supplement use. This would further assist policymakers in understanding consumer food choices and deliver targeted and effective messages to guide different subpopulations. Even though dietary supplements were used by more US residents from 2007 to 2018, we did not observe any improvement in reducing calcium deficiency. Considering the coexistence of excessive calcium intake and supplement use among a small proportion of the population, individuals in this category should be carefully guided on their dietary supplement use to achieve better public health outcomes.

Limitations should be noted when interpreting the results of this study. As a cross-sectional study, the trends reported here could be partially explained by the cohort effect. For example, the generation falling under the 2–8-year-old age group in the 1999–2000 survey cycle would become part of the 19–59-year-old age group in the 2017–2018 survey cycle. Insufficient calcium intake was determined in this study as a calcium intake level that failed to reach the EAR level. It should be emphasized that the EAR level, by definition, reflects the estimated median requirement for a defined group of persons [43]. The mean calcium intake level reported in the subgroup analysis should not be compared with RNIs, as most of the subgroups presented in this study would have mixed RNIs. The subgroup including other races and ethnicities was less represented and was composed of many different races and ethnicities, which made their trends less clearly presented in this study. Considering that this subgroup had the second highest rate of insufficient calcium intake across all race and ethnicity strata, further study is required to identify minor races and ethnicities of concern. This study focused on calcium consumption alone but did not look at the bioavailability of the calcium consumed. Different forms of calcium and other nutrients consumed such as magnesium, phosphate, vitamin D, and casein phosphopeptides could all influence the health outcomes of calcium consumption and should be analyzed and discussed in future studies [49,50,51,52,53,54,55,56,57]. 

## 5. Conclusions

This study provides a nationally representative estimate of the calcium intake level and its sources among the US population aged 2 years or older over a 20-year period (1999 to 2018). A large proportion of the US population had insufficient calcium intake, but trends in more recent years were more concerning. Different trends were observed among certain subgroups, highlighting the need for tailored guidance.

## Figures and Tables

**Figure 1 nutrients-16-00726-f001:**
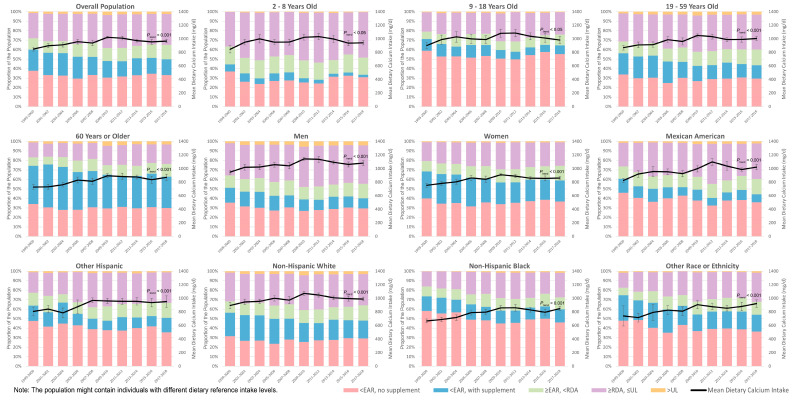
Dietary calcium intake and its distribution across DRIs among the US population, from 1999–2000 to 2017–2018.

**Figure 2 nutrients-16-00726-f002:**
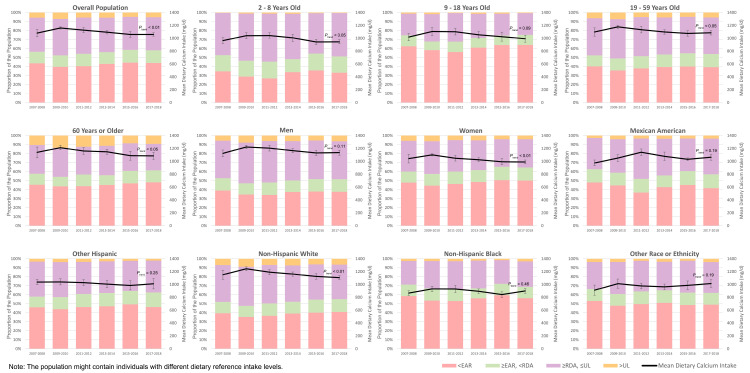
Calcium intake (dietary and supplement) and its distribution across DRIs among the US population, from 2007–2008 to 2017–2018.

**Table 1 nutrients-16-00726-t001:** Demographic overview of the sample persons included.

Characteristics	No. of Participants (Weighted %) by NHANES Cycle ^a^									
	1999–2000 (*n* = 7878)	2001–2002 (*n* = 8855)	2003–2004 (*n* = 8099)	2005–2006 (*n* = 8380)	2007–2008 (*n* = 8381)	2009–2010 (*n* = 8922)	2011–2012 (*n* = 7808)	2013–2014 (*n* = 7879)	2015–2016 (*n* = 7738)	2017–2018 (*n* = 6940)
Sex										
Men	3828 (48.6%)	4288 (48.7%)	3955 (49.1%)	4065 (48.8%)	4207 (47.9%)	4436 (48.7%)	3920 (49.4%)	3843 (49.3%)	3790 (48.7%)	3383 (48.7%)
Women	4050 (51.4%)	4567 (51.3%)	4144 (50.9%)	4315 (51.2%)	4174 (52.1%)	4486 (51.3%)	3888 (50.6%)	4036 (50.7%)	3948 (51.3%)	3557 (51.3%)
Age Group										
2–8 years	1113 (11.1%)	1399 (10.1%)	1175 (9.66%)	1372 (9.82%)	1379 (9.91%)	1419 (9.76%)	1415 (10.0%)	1189 (9.10%)	1157 (9.13%)	894 (8.90%)
9–18 years	2387 (14.7%)	2562 (15.3%)	2303 (15.0%)	2326 (14.6%)	1533 (14.1%)	1670 (14.2%)	1524 (13.3%)	1644 (13.8%)	1544 (13.6%)	1319 (13.6%)
19–59 years	2873 (57.6%)	3400 (58.8%)	2987 (57.5%)	3306 (57.4%)	3553 (58.4%)	3958 (57.6%)	3390 (57.4%)	3530 (57.0%)	3395 (56.0%)	2969 (55.4%)
≥60 years	1505 (16.6%)	1494 (15.8%)	1634 (17.9%)	1376 (18.2%)	1916 (17.6%)	1875 (18.5%)	1479 (19.4%)	1516 (20.1%)	1642 (21.2%)	1758 (22.1%)
Race and Ethnicity										
Mexican American	2665 (7.60%)	2218 (8.47%)	1985 (8.95%)	2204 (9.35%)	1686 (9.59%)	1941 (9.90%)	1052 (9.70%)	1334 (10.8%)	1468 (10.7%)	1011 (11.0%)
Other Hispanic	467 (7.58%)	398 (5.72%)	257 (3.30%)	269 (2.91%)	970 (5.37%)	945 (5.59%)	809 (6.66%)	745 (5.95%)	1007 (6.71%)	587 (6.88%)
Non-Hispanic White	2686 (67.9%)	3770 (69.5%)	3402 (70.5%)	3294 (69.6%)	3490 (67.7%)	3847 (65.9%)	2495 (63.3%)	2985 (62.8%)	2495 (61.3%)	2423 (59.3%)
Non-Hispanic Black	1784 (11.6%)	2133 (11.6%)	2119 (12.0%)	2207 (12.2%)	1878 (12.0%)	1657 (12.0%)	2167 (12.4%)	1697 (11.8%)	1653 (11.6%)	1600 (11.6%)
Other race or ethnicity ^b^	276 (5.39%)	336 (4.68%)	336 (5.17%)	406 (5.89%)	357 (5.34%)	532 (6.63%)	1285 (7.94%)	1118 (8.69%)	1115 (9.74%)	1319 (11.2%)
Pregnancy ^c^										
Pregnant	244 (5.58%)	261 (4.65%)	223 (4.76%)	315 (5.47%)	46 (2.25%)	48 (2.94%)	36 (2.15%)	43 (2.59%)	48 (3.06%)	38 (2.64%)
Not pregnant	492 (36.2%)	625 (39.7%)	675 (47.1%)	781 (50.3%)	702 (37.6%)	741 (35.2%)	542 (32.8%)	631 (35.6%)	621 (36.9%)	577 (39.2%)
Refused, do not know, or missing	1241 (58.2%)	1343 (55.7%)	1057 (48.1%)	1072 (44.2%)	1035 (60.1%)	1252 (61.9%)	1092 (65.1%)	1216 (61.8%)	1106 (60.1%)	898 (58.2%)
Lactation ^c^										
Currently breastfeeding	42 (1.60%)	61 (2.74%)	50 (2.45%)	49 (2.23%)	30 (1.65%)	32 (1.11%)	21 (1.58%)	40 (2.73%)	41 (2.36%)	38 (2.65%)
Not breastfeeding	155 (7.28%)	175 (6.54%)	115 (4.79%)	162 (6.12%)	99 (4.95%)	113 (5.01%)	69 (3.27%)	101 (6.87%)	118 (7.36%)	102 (8.80%)
Refused, do not know, or missing	1780 (91.1%)	1993 (90.7%)	1790 (92.8%)	1957 (91.6%)	1654 (93.4%)	1896 (93.9%)	1580 (95.2%)	1749 (90.4%)	1616 (90.3%)	1373 (88.6%)

Abbreviations: NHANES, National Health and Nutrition Examination Survey. ^a^ Percentages were adjusted for NHANES weights to generate nationally representative percentages; ^b^ included multiracial/ethnic groups; ^c^ among women aged 14 to 50 years.

**Table 2 nutrients-16-00726-t002:** Mean daily calcium intake among the US population, from 1999–2000 to 2017–2018.

	Mean Daily Dietary Calcium Intake (mg/d), (95% CI)									
	1999–2000	2001–2002	2003–2004	2005–2006	2007–2008	2009–2010	2011–2012	2013–2014	2015–2016	2017–2018
Total Diet										
Men	948 (919, 977)	1019 (977, 1061)	1024 (986, 1062)	1056 (1022, 1089)	1041 (996, 1085)	1141 (1116, 1165)	1135 (1093, 1177)	1091 (1066, 1116)	1060 (1024, 1096)	1079 (1044, 1114)
Women	753 (723, 782)	782 (755, 810)	804 (771, 837)	862 (824, 900)	843 (796, 890)	910 (896, 925)	888 (863, 913)	858 (837, 879)	856 (820, 891)	860 (834, 886)
2–8 years old	845 (791, 898)	950 (909, 991)	999 (938, 1061)	951 (913, 988)	951 (906, 995)	1022 (981, 1064)	1033 (989, 1076)	998 (943, 1052)	936 (896, 976)	941 (909, 973)
9–18 years old	902 (857, 947)	988 (920, 1057)	1028 (951, 1106)	999 (932, 1066)	989 (934, 1044)	1081 (1020, 1141)	1084 (1031, 1137)	1037 (987, 1088)	1012 (946, 1077)	981 (929, 1033)
19–59 years old	869 (834, 903)	910 (868, 952)	914 (874, 954)	987 (943, 1030)	961 (909, 1012)	1049 (1027, 1072)	1032 (996, 1068)	988 (959, 1016)	990 (951, 1029)	1005 (975, 1034)
≥60 years old	728 (689, 767)	730 (695, 765)	762 (728, 795)	831 (787, 876)	813 (768, 858)	894 (864, 925)	883 (835, 931)	876 (847, 905)	835 (784, 887)	872 (826, 918)
Mexican American	825 (793, 857)	920 (872, 967)	957 (885, 1029)	954 (913, 995)	925 (879, 971)	998 (951, 1046)	1098 (1050, 1145)	1038 (976, 1099)	986 (961, 1012)	1022 (985, 1059)
Other Hispanic	805 (740, 870)	842 (785, 899)	785 (714, 855)	875 (776, 975)	969 (933, 1006)	960 (922, 999)	955 (912, 998)	955 (900, 1011)	936 (873, 998)	951 (866, 1036)
Non-Hispanic White	894 (848, 941)	947 (906, 988)	955 (923, 986)	1001 (959, 1042)	972 (920, 1025)	1073 (1046, 1100)	1048 (1018, 1079)	1007 (980, 1034)	995 (957, 1032)	989 (961, 1017)
Non-Hispanic Black	666 (637, 695)	687 (658, 716)	716 (674, 759)	789 (752, 826)	796 (757, 836)	858 (825, 892)	860 (811, 908)	829 (793, 865)	794 (751, 837)	849 (816, 882)
Other race or ethnicity	740 (592, 888)	714 (649, 779)	791 (672, 910)	826 (763, 889)	809 (743, 875)	905 (846, 965)	877 (833, 921)	854 (812, 895)	879 (816, 942)	922 (867, 976)
Overall	848 (819, 876)	898 (868, 928)	912 (879, 945)	957 (923, 990)	938 (896, 980)	1022 (1009, 1036)	1010 (982, 1038)	973 (953, 993)	955 (920, 991)	966 (942, 991)
Diet and Dietary Supplements										
Men	NA *	NA *	NA *	NA *	1123 (1071, 1175)	1223 (1196, 1250)	1206 (1160, 1252)	1166 (1133, 1199)	1130 (1090, 1169)	1135 (1097, 1173)
Women	NA *	NA *	NA *	NA *	1042 (976, 1109)	1101 (1085, 1118)	1048 (1005, 1091)	1025 (996, 1055)	994 (945, 1042)	991 (960, 1021)
2–8 years old	NA *	NA *	NA *	NA *	965 (919, 1010)	1041 (995, 1087)	1044 (1000, 1087)	1008 (952, 1064)	943 (902, 984)	945 (914, 977)
9–18 years old	NA *	NA *	NA *	NA *	1019 (961, 1077)	1105 (1044, 1166)	1103 (1052, 1153)	1056 (1005, 1106)	1026 (956, 1096)	994 (940, 1047)
19–59 years old	NA *	NA *	NA *	NA *	1098 (1035, 1161)	1177 (1152, 1203)	1134 (1085, 1183)	1099 (1065, 1133)	1077 (1036, 1118)	1086 (1052, 1120)
≥60 years old	NA *	NA *	NA *	NA *	1139 (1055, 1223)	1214 (1179, 1249)	1161 (1107, 1215)	1148 (1107, 1189)	1088 (1019, 1156)	1086 (1030, 1141)
Mexican American	NA *	NA *	NA *	NA *	975 (930, 1021)	1051 (1001, 1101)	1141 (1090, 1192)	1078 (1013, 1144)	1032 (1010, 1053)	1064 (1020, 1108)
Other Hispanic	NA *	NA *	NA *	NA *	1038 (996, 1080)	1040 (993, 1087)	1028 (983, 1073)	1007 (950, 1063)	985 (915, 1056)	1011 (930, 1092)
Non-Hispanic White	NA *	NA *	NA *	NA *	1151 (1080, 1223)	1244 (1214, 1273)	1191 (1151, 1232)	1162 (1129, 1195)	1126 (1082, 1169)	1106 (1079, 1134)
Non-Hispanic Black	NA *	NA *	NA *	NA *	865 (828, 902)	930 (886, 974)	928 (873, 983)	893 (857, 930)	843 (802, 884)	899 (864, 935)
Other race or ethnicity	NA *	NA *	NA *	NA *	911 (831, 991)	1014 (944, 1084)	977 (934, 1021)	963 (920, 1007)	987 (919, 1055)	1014 (955, 1073)
Overall	NA *	NA *	NA*	NA *	1081 (1026, 1136)	1160 (1144, 1177)	1126 (1087, 1165)	1095 (1068, 1121)	1060 (1016, 1103)	1061 (1035, 1086)

Abbreviations: CI, confidence interval; NA, not applicable. The population might contain individuals with different dietary reference intake levels. * Data on calcium derived from dietary supplements was not collected.

**Table 3 nutrients-16-00726-t003:** Trends of calcium intake among the US population.

	From 1999–2000 to 2017–2018		From 1999–2000 to 2007–2008		From 2009–2010 to 2017–2018	
	Coefficient Estimate (mg/d·10 years^−1^)	*p* for Trend	Coefficient Estimate (mg/d·10 years^−1^)	*p* for Trend	Coefficient Estimate (mg/d·10 years^−1^)	*p* for Trend
Total Diet						
Men	62.5 (43.5, 81.5)	<0.001	110 (49.7, 170)	<0.001	−99.0 (−152, −46.5)	<0.001
Women	55.2 (38.8, 71.7)	<0.001	129 (67.5, 190)	<0.001	−66.6 (−104, −29.6)	<0.001
2–8 years old	34.7 (8.23, 61.2)	<0.05	108 (28.6, 188)	<0.01	−130 (−191, −69.3)	<0.001
9–18 years old	36.7 (4.46, 68.8)	<0.05	90.5 (3.97, 177)	<0.05	−135 (−226, −44.1)	<0.01
19–59 years old	70.5 (50.8, 90.3)	<0.001	129 (59.3, 199)	<0.001	−65.5 (−112, −19.5)	<0.01
≥60 years old	80.1 (56.5, 104)	<0.001	136 (69.3, 202)	<0.001	−44.9 (−113, 22.7)	0.19
Mexican American	88.1 (61.3, 115)	<0.001	105 (30.9, 179)	<0.01	−33.5 (−113, 45.6)	0.40
Other Hispanic	84.9 (42.9, 127)	<0.001	183 (97.6, 269)	<0.001	−18.9 (−123.0, 85)	0.72
Non-Hispanic White	51.0 (30.3, 71.7)	<0.001	104 (26.4, 181)	<0.01	−111 (−157, −64.8)	<0.001
Non-Hispanic Black	95.8 (75.4, 116)	<0.001	182 (124, 240)	<0.001	−42.6 (−100, 15.3)	0.15
Other race or ethnicity	95.3 (47.5, 143)	<0.001	123 (−51.9, 299)	0.16	31.3 (−57.4, 120)	0.48
Overall	59.0 (43.1, 75.0)	<0.001	118 (61.9, 175)	<0.001	−83.2 (−119, −46.9)	<0.001
Diet and Dietary Supplements						
Men	NA *	NA *	NA *	NA *	−126 (−182, −69.9)	<0.001
Women	NA *	NA *	NA *	NA *	−137 (−184, −89.8)	<0.001
2–8 years old	NA *	NA *	NA *	NA *	−147 (−210, −83.2)	<0.001
9–18 years old	NA *	NA *	NA *	NA *	−150 (−242, −57.4)	<0.01
19–59 years old	NA *	NA *	NA *	NA *	−120 (−173, −66.2)	<0.001
≥60 years old	NA *	NA *	NA *	NA *	−163 (−244, −82.4)	<0.001
Mexican American	NA *	NA *	NA *	NA *	−42.1 (−126, 42.0)	0.32
Other Hispanic	NA *	NA *	NA *	NA *	−48.6 (−152, 54.9)	0.35
Non-Hispanic White	NA *	NA *	NA *	NA *	−170 (−220, −121)	<0.001
Non-Hispanic Black	NA *	NA *	NA *	NA *	−73.8 (−142, −5.81)	<0.05
Other race or ethnicity	NA *	NA *	NA *	NA *	17.7 (−80.7, 116)	0.72
Overall	NA *	NA *	NA *	NA *	−132 (−175, −90.0)	<0.001

Abbreviation: NA, not applicable. The population might contain individuals with different dietary reference intake levels. * Results not available due to absence of data on calcium derived from dietary supplement from 1999–2000 to 2005–2006.

## Data Availability

Publicly available datasets were analyzed in this study. These data can be found here: https://www.cdc.gov/nchs/nhanes/index.htm (accessed on 2 May 2023).

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
