# Peer review of "Trends in Calcium Intake among the US Population: Results from the NHANES (1999–2018)"

_nutrients, 2024, doi:10.3390/nu16050726_

Round 1

Reviewer 1 Report

Comments and Suggestions for Authors

Trends in Calcium Intake among the US Population, Results 2 from the NHANES (1999-2018)

By Yu et al.

This paper concerns the intake of calcium in the US population, evaluated through a series of bi-annual surveys. The authors adequately presents the problem, the implications and methods, and to some extent also the data. The problem deserves attention, and for that matter the paper is a welcome contribution. However, The interpretation of the results appear to me to have great potential for improvement. I will be more specific below, but perhaps a consultancy with a professional statistician when it comes to data interpretations would benefit the manuscript.

1. The authors apparently use data from NHANES 1998-2018. However, when one look at the reference that the Authors give (ref 7), is is just a website, that doesn't link directly to reports on the NHANES surveys. I noticed however, that the age grouping in the linked NHANES site is different from what the authors use when it comes to minors: The Tables in the document that the authors link to has the groupings of 2-5, 6-11 and 12-19 years of age, while the authors use 2-8 and 9-18 years of age. Please explain, and provide relevant references. The only other reference that mentions NHANES (ref 9) uses yet another grouping.

2. Are the NHANES data really representative, as the authors claim in line 108? The justification is a reference to a NCHS site with tips for computations on specific software, which can hardly serve as scientific justification. I am aware that NHANES works with sampling weights in order to obtain estimated national averages (which is also noted in a footnote to Table 1), and in fact the age groups in the link that the authors supplied indicate age groups of more or less equal sizes, despite them ranging over very a different number of age years. So a more precise statement on the representativity of exactly these data seems to be in place.

3. In the discussions section, that authors almost exclusively deals with what has happened between the 2009-10 survey and the 2017-2018 survey (termed "2009" and "2018" in the paper). But the data material extends for twice that period? Apart from a comparisons between non-hispanic whites and mexicans for 1999 to 2008 and their changes in calcium levels from start to end, I can see no reference to the first part of the data and the development there. I noted that the authors write that an increasing trend in calcium intake had just reported by 2009-10 (line 242), but that is the reference 7 which is just an imprecise reference to data tables and thus subjective interpretation, and not a published scientific conclusion. Even so, it hardly justifies in my opinion ignoring half of the data material. Then why show it at all? I suggest that the authors either limit the data material to what they want to discuss, or alternatively discuss the data that the present. It also bothers me somewhat that cutting off the data at 2009 is the optimal way of adapting the data to the message of the authors. Prior to that, a positive development seem to take place, and table 3 reveals that all subgroups have significantly increased their calcium intake over the survey period, telling the opposite story than what the authors write based on the 2009-2018 data. The cut point of 2009 should be justified.

4. In lines 197-198, the authors report that the 9-18 year olds had the consistently highest fraction with intake lower than their EAR among the age groups. On reference to the supplementary data, this appears incorrect to me; in the supplementary data it is seen that it is only in the very last of the 10 NHANES survey that 9-19 years olds are reported with the highest proportion; in all the other 9 surveys, the 69+ group have higher fractions. 

5. In lines 199-200, the authors write that Nin-Hispanic Black followed by other race have the highest proportion with insufficient intake. That must be among the racial groups only, as the black group has a consistently lower proportion when compared to the 60+ group and to some extent the 9-18 group.

6. In the start of the discussion, the authors mention that they considered strata by age group, sex and race/ethnicity. However, women are not mentioned by a single word in the discussion, despite them being consistently more problematic than men, as the authors do note in their result section. Why is it so, and does it have any implications for targeted interventions? The authors do write in line 246 about support for tailored decisions from politicians.

7. The groups with a consistently higher proportion than the overall on intake below their EAR are old people, teenagers, black people, women and 'other races'. This is also noted by the authors in the results section. According to appendix B, at least for age and gender, this more or less corresponds to those having an EAR of 1000+mg. Do the authors agree with this and in that case, what are the implications? One could imagine increased problems if people do not get their calcium when they really need it. This could also be discussed in terms of the 'tailored interventions' that the authors write about.

8. I miss a discussion of the racial aspect that the authors have chosen to include. Is the heterogeneity presumed to be due to genetics? Socioeconomy (ie. poverty etc)? Both? Culturally founded? etc etc.

9. In lines 229-231, the authors write: "The consistent low total 229 calcium intake among non-Hispanic Black could be explained by their low dietary calcium 230 intake and calcium-containing supplement usage rate." Isn't that a circular argument? Ie. they have low calcium rates because they don't eat any?

10. In lines 233, the authors write that calcium intake significantly decreased for whites between 2009 and 2018, but not for mexicans. In fact, it didn't decrease significantly for any racial group apart from the whites according to your Table 3. So why single out the mexicans? I am aware that you write above that they have a similar progression earlier, but it is not much different from the blacks and other races/ethnicity, only hispanics seem to have a slightly different development, with a dive in the early part.

11. Line 237: The authors write that the 'sharp decrease' in dietary calcium intake for whites should be noted. Which sharp decrease? I only see a change in decrease from 2009-10 to 2011-12, after that the decrease is constantly the same, and the early decrease is shared with many other subgroups, eg. women, adults, and the overall population.

Author Response

Thank you for your in-depth review and suggestions to improve the manuscript! Please see our response to your comments in the attached file, with your original comments in black and our response in blue.

Reviewer 2 Report

Comments and Suggestions for Authors

This paper described trends in calcium intake among the US population from 1999 – 2018, stratified across age groups, sex, race, and ethnicity with and without dietary supplements. Data was from NHANES 1999-2000 to 2017-2018.

The authors state that a large percentage of the population still suffers from insufficient calcium intake (lines 275-277). I found it difficult to follow this with the current organization of the data. Appendix B, Table B1, lists the DRIs for calcium by life stage (mg/day). Both the EAR, RDA, and UL are provided. None of the figures or tables follow the age groups provided in Table B1. It isn’t easy to arrive at the researcher’s conclusion based on the current organization of the data. For example, the age groups in the data tables are 2-8, 9-18, 19-59, and >60. The DRIs are based on 2-3, 4-8, 9-18, 19-50 years old. So how much should the 2-8 group consume, 700 or 100o mgs?

The researchers report concerning trends and state the US population still consumes inadequate amounts of calcium. There seem to be inconsistencies in what the data shows. Table B1 shows the RDA for 19–50 year-olds to be 1000 mg/day, but Table 2, 2017-2018, shows the age group 19-59 consumed 1005 mg/day. This amount is above the RDA, suggesting that this group has no problem with calcium intake. From 2009, data was collected five times; three of the four times were >1000, and two were only 10-12 mg less than 1000mg.

The calcium intake of 9-18 year olds decreased the most (lines 153-154). Why was this? What foods decreased to account for the lower calcium intake?

Figures 1 and 2 show the calcium intake and what percentage met the EAR and RDA or exceeded the upper limit. The legend describing each group is too small and should be increased so it is legible. The y-axis is not labeled. A line for the RDA or EAR should be added.

Where is Table 3 mentioned in the text?

Author Response

(The authors gave the same response as above.)

Reviewer 3 Report

Comments and Suggestions for Authors

The abstract needs editing.:

line 12: population have been;...... We used data from NHANES

line 14: Among the 80,880

line 15: included, in our study, a

line 20: children and non-Hispanic 

line 23: calcium intake were observed

Introduction: 

line 49: 2014, but the study did not

line 50: dietary calcium intake used

line 56: strategies that could lead 

Material and Methods

line 63: data collection were described 

line 70: factors. To allow...

line 74: or above {8,30,31]. 

line 75: sample persons were divided

Results:

" with extreme energy intake": This needs to be defined and included first as part of the methods and then again here

Author Response

(The authors gave the same response as above.)

Round 2

Reviewer 1 Report

Comments and Suggestions for Authors

The authors have addressed my concerns. However, I will still suggest to make reference 7 more precise, as it is just a reference to something that is present on the internet, without being specific.

Perhaps the link

https://www.ncbi.nlm.nih.gov/books/NBK589560/

will work. Note however that it is published in 2014 and not 2010.

Apart from this I have no further issues. Good luck with the publication.

Author Response

We appreciate your time and effort. Please find attached our point-to-point response to your comments.

Reviewer 2 Report

Comments and Suggestions for Authors

The authors have provided a lengthy response to my comments and suggestions. They state that they selected the age groups are based on the Dietary Guidelines. The age groups used by the Dietary Guidelines (DG) are shown on page 133 (page 146 in pdf viewer) of reference 8 and re based on the RDA. The age groups are 2-3, 4-8, 9-13, 14-18, 19-30, 31-50, and 51+. The authors have consolidated the groups from 7 to 4 because it would take “extra amount of words and space…to present such so many subgroups.” This reason is inappropriate. Since the RDA is the basis for the calcium recommendations in the Dietary Guidelines, the age groups in this paper should jib with the age groups used in the DG since that is what they claim they used: 2-3 and 4-8 condensed to 2-8; 9-13 and 14-18 condensed to 9-18; and 19-50 and 51+.

Author Response

We appreciate your time and effort. Please find attached our response to your comments.
